# Hybrid Reconstruction Approach for Polychromatic Computed Tomography in Highly Limited-Data Scenarios

**DOI:** 10.3390/s24216782

**Published:** 2024-10-22

**Authors:** Alessandro Piol, Daniel Sanderson, Carlos F. del Cerro, Antonio Lorente-Mur, Manuel Desco, Mónica Abella

**Affiliations:** 1Bioengineering Department, Universidad Carlos III de Madrid, 28911 Leganes, Spain or alessandro.piol@unibs.it (A.P.); dsanders@ing.uc3m.es (D.S.); carlosfe@pa.uc3m.es (C.F.d.C.); alorente@ing.uc3m.es (A.L.-M.); 2Department of Information Engineering, University of Brescia, Via Branze, 38, 25123 Brescia, Italy; 3Instituto de Investigación Sanitaria Gregorio Marañón, 28007 Madrid, Spain; 4Centro Nacional de Investigaciones Cardiovasculares Carlos III (CNIC), 28029 Madrid, Spain; 5Centro de Investigación Biomédica en Red de Salud Mental (CIBERSAM), 28029 Madrid, Spain

**Keywords:** artifact removal, beam hardening, CT, deep learning, limited data, L2-PICCS

## Abstract

Conventional strategies aimed at mitigating beam-hardening artifacts in computed tomography (CT) can be categorized into two main approaches: (1) postprocessing following conventional reconstruction and (2) iterative reconstruction incorporating a beam-hardening model. While the former fails in low-dose and/or limited-data cases, the latter substantially increases computational cost. Although deep learning-based methods have been proposed for several cases of limited-data CT, few works in the literature have dealt with beam-hardening artifacts, and none have addressed the problems caused by randomly selected projections and a highly limited span. We propose the deep learning-based prior image constrained (PICDL) framework, a hybrid method used to yield CT images free from beam-hardening artifacts in different limited-data scenarios based on the combination of a modified version of the Prior Image Constrained Compressed Sensing (PICCS) algorithm that incorporates the L2 norm (L2-PICCS) with a prior image generated from a preliminary FDK reconstruction with a deep learning (DL) algorithm. The model is based on a modification of the U-Net architecture, incorporating ResNet-34 as a replacement of the original encoder. Evaluation with rodent head studies in a small-animal CT scanner showed that the proposed method was able to correct beam-hardening artifacts, recover patient contours, and compensate streak and deformation artifacts in scenarios with a limited span and a limited number of projections randomly selected. Hallucinations present in the prior image caused by the deep learning model were eliminated, while the target information was effectively recovered by the L2-PICCS algorithm.

## 1. Introduction

The origin of the beam-hardening effect lies in the polychromatic nature of X-ray sources, whereby the mean energy increases its value when traversing a material since low-energy photons are more easily absorbed than high-energy photons. The beam-hardening effect leads to two distinct types of artifacts in CT reconstructed images: cupping in homogeneous regions and dark bands among dense areas in heterogeneous regions [1], compromising the quantitative fidelity of the image.

We can find multiple beam-hardening correction approaches in the literature. It is common to pre-harden the beam by using a physical filter that eliminates most of the low-energy photons [1]. However, this method falls short of completely suppressing artifacts, making additional image processing techniques necessary. The processing method implemented in most scanners is water linearization, which assumes that the sample is homogeneous and therefore only addresses cupping artifacts [2]. To correct both cupping and dark bands, some works model the beam-hardening effect using knowledge about the X-ray spectrum together with an estimation of the tissue thicknesses traversed [3,4]. The need for spectrum knowledge can be overcome by using a beam-hardening model based on information from a calibration phantom [5]. Other approaches, such as maximizing the flatness [6] or the entropy [7] in reconstructed images, avoid the need for explicit beam-hardening model characterization. However, these methods still require image segmentation, a challenging task in limited-data acquisitions where streaks and edge distortions compromise segmentation accuracy. In such scenarios, iterative algorithms can enhance segmented masks in successive iterations. The work proposed in [8] included a polychromatic model of the source but required knowledge of the spectrum to incorporate the energy effect into the projection matrix. This requirement was eliminated in [9], based on a simplified polychromatic model that uses two parameters together with the calibration step of the water linearization method. The polychromatic model was further improved in [10] by using a calibration step to model the beam-hardening function. While these methods can be implemented for the correction of limited-data CT with beam hardening, they have two main downsides: 1) applying a polychromatic model is computationally expensive and relies on heuristic approximations of the tissues, and 2) images in extreme scenarios with a very low number of projections are significantly distorted compared to the target images.

The computational cost associated with tomographic reconstruction can be significantly mitigated by the use of DL methods, but few works in the literature have dealt with beam-hardening artifacts. Li-ping et al. [11] exploited a three-layer convolutional network trained with simulated reconstructed data to correct industrial CT images made up of a single material, which is an easier problem than dealing with medical images. Ji et al. [12] followed the workflow of the empirical beam-hardening correction method proposed in [6] by replacing the empirical terms with a neural network, although the method showed a loss of spatial resolution for the low-dose cases. In [13], the authors proposed an end-to-end workflow that reconstructs images free of beam-hardening artifacts from the original projections based on the concatenation of three consecutive U-Net [14] networks, but the evaluation was only performed on images of 64 × 64 pixels—a matrix size far from those of real cases—and in most cases, the images resulting from their method did not outperform the FDK reconstruction. None of these works address the additional artifacts that may appear simultaneously, such as those caused by limited projections or a limited span.

Most DL works on tomographic image reconstruction have focused on correcting artifacts that are generated in scenarios of limited projections and limited span separately. In scenarios with a limited number of projections (LNP), most approaches aim to improve the sinograms [15], the image obtained from an initial FDK reconstruction [16], or both the sinogram and the reconstructed image [17,18]. Nevertheless, these approaches present some significant loss of details when the number of projections is too small. Similarly, to solve the issues arising from a limited span angle (LSA), some works focus on improving the image after an FDK reconstruction [19,20], the image in the wavelet domain [21], or both the sinogram and the reconstructed image [22]. Nevertheless, none of these works address simultaneously the artifacts caused by randomly selected projections in a gating scenario or by a highly limited angle span. Also, when evaluated for a low number of projections, these methods have shown significant artifacts. Additionally, scenarios where the span angle and the number of projections differ from those used during training have not been evaluated. Finally, an important problem of relying completely on DL to obtain the final reconstruction is the risk of generating false structures known as hallucinations [23], especially in the regions that are most affected by the limited data artifacts.

These limitations have been mostly addressed with the advent of plug-and-play [24,25] and unrolling [26] algorithms which embed a DL model within an iterative scheme. The former can be challenging to implement in cases with a very limited number of projections due to the domain shift between the training data and the intermediate reconstructions. The latter can be computationally prohibitive for 3D problems such as cone beam reconstruction due to the need to implement the projection and backprojection steps within the neural network. Additionally, it is unclear if it is feasible to incorporate a beam-hardening correction within this framework as it would require a polychromatic projection model in both cases.

An intermediate approach between full reconstruction with DL and the embedding of a DL model within an iterative algorithm has been to produce a DL reconstruction to use as a prior image within an iterative algorithm [27,28]. While this approach incorporates DL less elaborately compared to unrolling algorithms or plug-and-play algorithms, it circumvents some of the practical problems faced by those algorithms. Following this idea, we propose PICDL, a hybrid reconstruction method based on the combination of a modified version of the original PICCS algorithm [29,30], which we name L2-PICCS, with a DL model to generate a prior image to compensate artifacts due to beam hardening, applicable to situations with a limited number of randomly distributed projections and instances with restricted span data. L2-PICCS substitutes the prior TV regularization term with the L2 norm, making PICCS robust to very-low-dose scenarios by minimizing the appearance of streaks in the final reconstruction. The DL model has been tested in scenarios where the projections do not match those of the training data for either standard dose (SD), low dose (LD), LNP, or LSA scenarios. While L2-PICCS does not use a polychromatic projection model, the results show that the simple correction of the beam-hardening within the prior image allows for the correction of beam-hardening in the result of the algorithm. The effectiveness of the algorithm was tested on real small-animal cone beam CT data.

## 2. Methods

PICDL is a hybrid method that consists of three stages: (1) the preliminary FDK reconstruction with artifacts, which is the input to the network, (2) the generation of a prior image from this preliminary reconstruction with a DL architecture (DeepBH) that works slice by slice and is trained to correct beam-hardening along with either SD, LD, LSA, or LNP, and (3) the use of this prior image in the L2-PICCS algorithm to obtain the final reconstruction without artifacts (Figure 1).

The following sections briefly describe L2-PICCS for completeness and describe the architecture of DeepBH used to generate the prior image.

### 2.1. L2-PICCS Method

Assuming that the solution *u* has a sparse gradient, and that it is close to a prior image *u_p_*, PICCS finds *u* as the solution to the optimization problem [30]:(1)$$\underset{u}{\min}\left(1-\alpha \right)T_{1}\left(u\right)+\alpha T_{2}\left(u-u_{p}\right),s.t\;{\left\|Fu-f\right\|}_{2}^{2}\leq \sigma^{2},\;u\geq \;0$$
where f is the limited projection data, F is the forward operator, $T_{1}$ and $T_{2}$ can be any transform previously used in Compressed Sensing (CS) studies, *σ* represents the variance of the noise in the data, and *α* weights the contributions of $T_{1}$ and $T_{2}$. We add the positivity constraint u≥0 as in [31].

While in [30] the authors use the total variation pseudo-norm for both $T_{1}$ and $T_{2}$, L2-PICCS differs from this original PICCS implementation by using the L2 norm as the second transform $T_{2}$. This modification is carried out to avoid transferring into the solution the high gradient streaks present in the $\left(u-u_{p}\right)$ regularization term, caused by the very low number of projections in our data.

### 2.2. Split Bregman Algorithm

Equation (1) can be solved by using the Split Bregman method [32]. We introduce the Bregman iterators $b_{x}^{k}$, $b_{y}^{k}$, and $b_{v}^{k}$, and the auxiliary variables $d_{x},\;d_{y},\;v$ under the constrains ${d_{y}=\nabla }_{x}u,\;{d_{y}=\nabla }_{y}u,\;v=u$. By adding the constraints through penalty functions, we obtain the following minimization problem:(2)$$\begin{array}{c} \underset{u,d_{x},d_{y},v}{min}\left(1-\alpha \right){\left\|\left(d_{x},d_{y}\right)\right\|}_{1}+\alpha {\left\|u-u_{p}\right\|}_{2}+\Phi \left(v_{i}\geq 0\right)+\frac{\mu }{2}{\left\|Fu-f\right\|}_{2}^{2}\\ +\frac{\lambda }{2}{\left\|d_{x}-\nabla_{x}u-b_{x}^{k}\right\|}_{2}^{2}+\frac{\lambda }{2}{\left\|d_{y}-\nabla_{y}u-b_{y}^{k}\right\|}_{2}^{2}+\frac{\gamma }{2}{\left\|v-u-b_{v}^{k}\right\|}_{2}^{2} \end{array}$$
where Φ(v>0) is the indicator function for the positivity constraint, and *λ*, *µ*, and *γ* are the penalty function weights.

We decouple the problem by alternating between the optimization of the terms using the L2 norm, L1 norm, and the indicator function of positive values. The first is solved by setting the gradients to 0, leading to the following system of equations that we iteratively compute with the Krylov solver:$$Ku^{k+1}=r^{k}$$
(3)$$K=\mu F^{T}F+\lambda {D_{x}}^{T}D_{x}+\lambda {D_{y}}^{T}D_{y}+\left(2\alpha +\gamma \right)\;I$$
$$r^{k}=\mu F^{T}f^{k}+2\alpha \;u_{p}+\lambda {D_{x}}^{T}\left(d_{x}^{k}-b_{x}^{k}\right)+\lambda {D_{y}}^{T}\left(d_{y}^{k}-b_{y}^{k}\right)+\gamma \left(v^{k}-b_{v}^{k}\right)$$

The L1 subproblem is solved using the proximal operator of the L1 norm, the general shrinkage formula, applied to the variables $d_{x},d_{y},v$. The positivity constraint is enforced using the projector (i.e., the proximal operator of the indicator function) onto the set of positive values.
$${d_{x}}^{k+1},{d_{y}}^{k+1}=max\left(s^{k}-\left(1-\alpha \right)/\lambda ,0\right)\;\frac{\left|D_{j}u^{k+1}+b_{j}^{k}\right|}{s^{k}},\;j=x,y$$
(4)$$s^{k}=\sqrt{{\left|D_{x}u^{k+1}+b_{x}^{k}\right|}^{2}+{\left|D_{y}u^{k+1}+b_{y}^{k}\right|}^{2}}$$
$$v^{k+1}=max(u^{k+1}+b_{v}^{k},0)$$

The Bregman iterators are updated as follows:$${b_{x}}^{k+1}={b_{x}}^{k}+\nabla_{x}u^{k+1}-{d_{x}}^{k+1}$$
(5)$${b_{y}}^{k+1}={b_{y}}^{k}+\nabla_{y}u^{k+1}-{d_{y}}^{k+1}$$
$${b_{v}}^{k+1}={b_{v}}^{k}+u^{k+1}-v^{k+1}$$
$$f^{k+1}=f^{k}+f-Fu^{k+1}$$

The resulting hyperparameters are *μ*, that weights the contribution of the limited projection data f, *α* and *λ* that weigh the regularization terms, and *γ* that regulates the convergence speed of the algorithm. The differences in L2-PICCS with respect to the original PICCS are reflected in the removal of the proximal operator originally associated with the $T_{2}$ transform from the L1 subproblem (Equation (4)) and the incorporation of the L2 norm into the Krylov solver (Equation (3)).

### 2.3. DeepBH

The prior image is obtained with DeepBH, a DL architecture based on a modification of the U-Net architecture [14] replacing the original encoder with ResNet-34 [33] to take advantage of the improved performance offered by the residual block. The decoder mainly comprises oversampling blocks that perform subpixel convolution via pixel shuffling with CNN resize initialization (ICNR) [34], followed by two blocks consisting of a convolutional layer and a rectified linear unit (ReLU). Each oversampling block is linked to its corresponding encoder block via skip connections. The complete architecture is depicted in Figure 2.

We trained the model using a Smooth L1 loss function, defined as follows:(6)$$l\left(x,\;u\right)=\frac{1}{N}\;\sum_{n=0}^{N}l_{n}(x_{n},\;u_{n}),\;l_{n}(x_{n},\;u_{n})=\left\{\begin{array}{c} 0.5\times {\left(x_{n}-u_{n}\right)}^{2}\;if\;\left|x_{n}-u_{n}\right|\;\leq 1\\ \left|x_{n}-u_{n}\right|-0.5>1 \end{array}\right.\;$$
where $x={\left[x_{1},\ldots ,x_{N}\right]}^{T}$ is the predicted image, $u={\left[u_{1},\ldots ,u_{N}\right]}^{T}$ is the target image, and N is the number of images present in the dataset. This loss function combines the advantages of L1 loss, i.e., stable gradients when the difference between the prediction and the target is large, and L2 loss, i.e., fewer oscillations during updates, when the differences between prediction and target are small. We used the Adam optimizer [35] due to its higher convergence speed, with a weight decay equal to 10^−2^ as a regularization strategy. The layers of the CNN were initialized using the Kaiming uniform method [36]. Fine-tuning was employed as a training strategy. We first trained only the encoder until convergence with a learning rate of 1 × 10^−4^ determined by the Leslie N. Smith test [37]. Subsequently, the model was trained end-to-end with a learning rate of 1 × 10^−7^ to improve the model performance. To ensure the reproducibility of our training results, we initialized the weights and biases of the CNN using a specified random seed.

To generate the prior image, the model was trained at half the size to reduce the risk of transferring potential hallucinations from the network’s output to the final image reconstructed with PICDL. To this end, input images were downsampled to half size by averaging the neighborhoods of 4 pixels and the output image was upsampled by bilinear interpolation.

## 3. Experiments and Results

We first used DeepBH directly in two conventional scenarios, standard dose and low dose, to evaluate the performance and robustness of the model used to obtain the prior image. In the second step, we evaluated the whole hybrid method PICDL in the scenarios in which a simple postprocessing like DeepBH is not enough, i.e., in highly limited-data scenarios, either with a limited span angle or limited number of projections. L2-PICCS has been implemented only in the central slice of the volume to simplify the projection and backprojection steps in the iterative algorithm. The results of PICDL and DeepBH were compared with those of SART-PICCS, which was adapted from [38] by changing ART by SART given that they are equivalent and SART is faster.

### 3.1. Datasets

We used 11 rodent head studies acquired with the CT subsystem of an ARGUS/CT system [39], a cone-beam micro-CT scanner based on a flat panel detector with a source-detector distance of 370.95 mm. We obtained 360 projections within a span angle of 360 degrees, with a projection size of 516 × 574 pixels and a pixel size of 0.2 mm. Experiments were carried out in accordance with the Animal Experimentation Ethics Committee of the Community of Madrid (Ref. PROEX 332/15), following the EU Directive 2010/63EU and Recommendation 2007/526/EC, and the enforcement in Spain from RD 53/2013.

We trained and validated one model for each of the following four data scenarios generated from the acquired data of the 11 rodent studies (Figure 3):Standard dose (SD) scenario. Complete data, that is, 360 projections with a span angle of 360 degrees, resulting in seven studies for training, two for validation, and two for test.Low-dose (LD) scenario. We selected every second projection from each study (180 projections covering a 360-degree span), resulting in seven studies for training, two for validation, and two for test.Limited span angle (LSA) scenario. This scenario entailed the random selection of three span angles between 90 and 160 degrees for each of the 11 rodent studies. Consequently, we obtained a total of 21 studies for training, 6 for validation, and an additional 6 for testing.Limited number of random projections (LNP) scenario. We conducted three rounds of selection, randomly choosing a varying number of projections between 30 and 60 within a 360-degree span for each of the 11 datasets. This process yielded a total of 21 studies allocated for training, 6 for validation, and another 6 for testing.

Table 1 shows the resulting number of 2D images for each scenario.

Preliminary reconstructions consisted of volumes of size 448 × 512 × 496, with a voxel size of 0.122 × 0.122 × 0.122 mm^3^, obtained with an FDK-based method implemented in FuxSim [40]. Target images free from beam-hardening artifacts were obtained from the SD datasets using FDK + 2DCalBH [5]. Since the prior volume in PICCS is used to recover the texture and help recover the main structures, the CTs were downsampled by half in the three dimensions, resulting in 224 × 256 × 248 volumes. The prior image obtained with the DL model was then rescaled to 516 × 516 × 496 to be used in the PICCS algorithm.

### 3.2. Parameters of Iterative Methods and Evaluation Metrics

Table 2 shows the parameters used for the L2-PICCS algorithm, which were selected to maximize both data consistency and perceptual appearance after performing a visual grid search.

The robustness of PICDL against different priors obtained from different datasets was assessed by training the model for the LNP case with two different random seeds (33 and 42).

SART-PICCS was run for 40 global iterations that include 10 PICCS iterations and 1 SART iteration. We used a PICCS step size of 0.2, a SART step size of 1.0, a prior weight of 0.35, and a relax value of 0.01, under all scenarios.

The performance on the test studies was assessed visually and quantitatively using Peak Signal-to-Noise Ratio (PSNR) to quantify texture and pixel value differences, Structural Similarity Index (SSIM) to quantify structural differences with respect to the target images [41], and Correlation Coefficient (CC) [42] to measure the overall linear similarity between the images, providing a comprehensive evaluation of both pixel-wise and structural congruence.

### 3.3. Results in Conventional Scenarios

Results for the SD and LD scenarios showed that a postprocessing step with DeepBH was able to correct both streaks and dark band artifacts, with a result very similar to the target regardless of the dose (Figure 4). This is quantitatively supported by the improvement in PSNR and SSIM values, especially for the LD scenarios due to the elimination of streaks (Figure 5).

### 3.4. Results in Highly Limited-Data Scenarios

Figure 6 shows that SART-PICCS is sensitive to the hallucinations in the prior image, while PICDL achieves almost identical results for priors obtained with different random seeds, with only minor differences appearing in internal structures with higher intensity, reaching very similar PSNR and SSIM values (Table 3). This demonstrates the robustness of the method against changes in the prior image.

Figure 7 shows that DeepBH, SART-PICCS, and PICDL were able to correct the deformations caused by the reduced span angle (see white arrows in Figure 7A and Figure 7F), leading to PSNR and SSIM improvements with PICDL as high as 100% and 67%, respectively (Table 4). While SART-PICCS introduced blurring in the reconstructed image owing to the downsampled prior, PICDL could recover high-resolution edges.

In the case of the LNP scenario, while we see an elimination of beam-hardening artifacts and streaks with all methods, DeepBH showed some hallucinations in the form of new or incomplete structures (see white arrows Figure 8B and Figure 8G). While SART-PICCS transferred the hallucinations of the prior image into the reconstructed image, PICDL was able to avoid them (Figure 8E and Figure 8J), resulting in an increased PSNR and SSIM of more than 86% and 121%, respectively, compared to FDK.

## 4. Discussion and Conclusions

This paper presents a hybrid reconstruction method designed to mitigate beam-hardening artifacts in CT images. This method is valid for limited-data scenarios and is based on the combination of L2-PICCS, a modified version of the PICCS algorithm, with DeepBH, a DL architecture that generates the prior information. While this prior information is available in dynamic acquisitions as respiratory gating, the proposed method enables the use of L2-PICCS for the limited-data cases where complete data are not available (i.e., static low-dose acquisitions).

Evaluation with rodent head studies in a CT scanner showed that DeepBH was able to correct beam-hardening artifacts and streaks in scenarios with standard and low-dose data without introducing hallucinations. In both highly limited-data scenarios, LSA, and LNP, DeepBH was able to eliminate beam-hardening, streaks, and deformation artifacts arising due to the lack of projections and improve the texture, especially in soft tissue. Nevertheless, in LNP, the lack of data in internal structures and surface edges resulted in the generation of hallucinations not present in the other scenarios. This suggests that edge deformations are more easily learned by the network than small structures hindered by severe streaks. It is difficult to assess the diagnostic relevance of these hallucinations as it depends on the specific pathology or purpose of the study, but they introduce an uncertainty factor that precludes its use for clinical applications and hampers a possible use in preclinical scenarios depending on the specific application.

The appearance of hallucinations under the LNP scenario justifies the need for hybrid techniques such as PICDL to properly reconstruct data with a highly limited number of projections and recover internal structures and edges. The results show that while SART-PICCS was not able to correct small hallucinations in the prior image, the incorporation of the original measured data through our hybrid approach effectively eliminates the hallucinations introduced by the DL model into the prior image. The use of Split Bregman in our implementation allows the PICCS problem to be solved accurately, while SART-PICCS approximates the gradient of the TV norm by means of a relax parameter to apply gradient descent, what may cause its inability to correct small hallucinations in the prior image, leading to an incorrect reconstruction of small gradients.

The evaluation of PICDL using different priors, resulting from different initializations of the weightings and biases of the CNN, supported the robustness of the proposed method since it can obtain results that are visually and quantitatively very similar to each other.

The main challenge with PICDL was the selection of the regularization parameters for L2-PICCS, which was carried out heuristically depending on the type of artifact, the number of total projections, and their spatial distribution within the total span angle. Future work will explore the possibility of optimizing these parameters with a deep learning model, provided that enough complete PICDL studies are available. Other methods such as the so-called unrolling methods include the DL model as part of the iterative method [43,44]. Nevertheless, the use of unrolling methods with 3D data is limited by the large computational memory costs.

One limitation of our study is the reduced size of the training database. Future work will require the creation of larger datasets and the evaluation of the method on anatomical regions other than rodent head studies to determine the need to independently train the DL model on each body part.

While PICDL does not include a polychromatic projector model, it shows that the use of a beam-hardening corrected prior allows for beam hardening to be solved in a way that is not computationally demanding.

The proposed method is an easy-to-implement approach to incorporate the potential of the deep learning strategies to limited-data CT, preventing the transfer of hallucinations to the final solution. This could be especially useful in settings with mechanical limitations, as is the use of C-arms to obtain tomographic images in the operating room, or where the radiation dose must be reduced, as in screening.

In conclusion, PICDL has demonstrated effectiveness in compensating for beam-hardening artifacts across various scenarios, successfully correcting streaks and deformation artifacts present in highly limited data, paving the way for real-time quantitative tomography in situations involving non-standard trajectories, such as those encountered in the used C-arm systems during surgery.

## Figures and Tables

**Figure 1 sensors-24-06782-f001:**
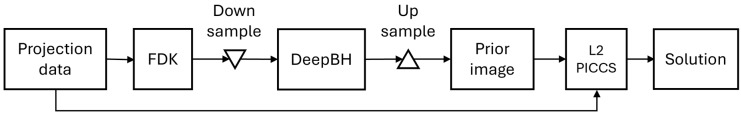
Flowchart of the proposed method.

**Figure 2 sensors-24-06782-f002:**
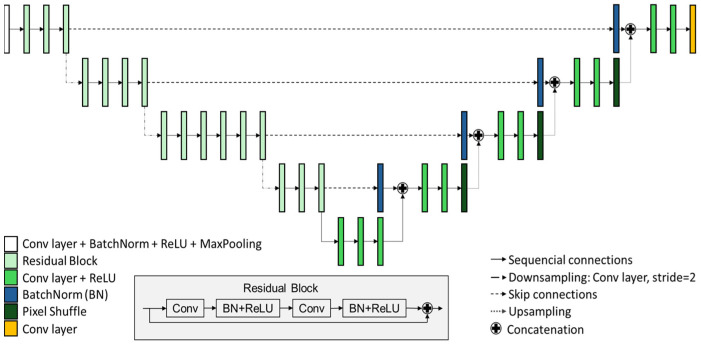
Proposed U-Net network architecture.

**Figure 3 sensors-24-06782-f003:**
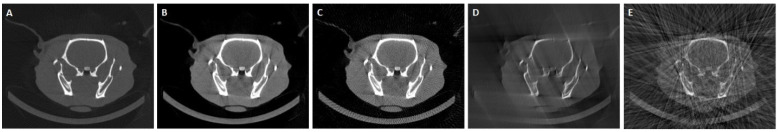
Central axial slice of the reconstructions of one of the tests cases for the target (**A**) and SD (**B**), LD (**C**), LSA (**D**), and LNP (**E**) scenarios.

**Figure 4 sensors-24-06782-f004:**
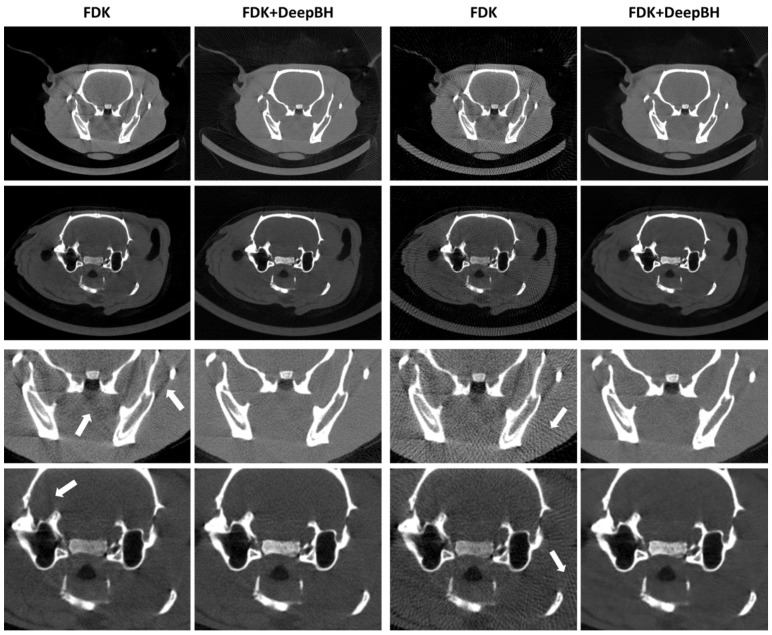
Top: central axial slices of the FDK reconstructions scenarios and DeepBH results for SD and LD scenarios for the two test studies. Bottom: zoomed-in images. Arrows in zoomed-in images point to beam hardening (first column) and streaks (third column).

**Figure 5 sensors-24-06782-f005:**
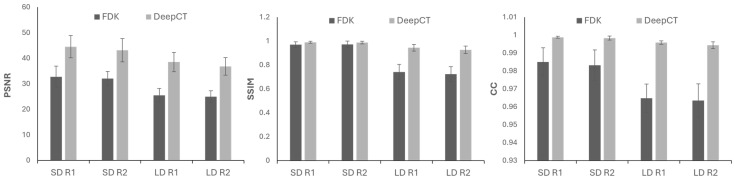
Mean and standard deviation of PSNR, SSIM, and CC values calculated for the SD and LD scenarios in each slice.

**Figure 6 sensors-24-06782-f006:**
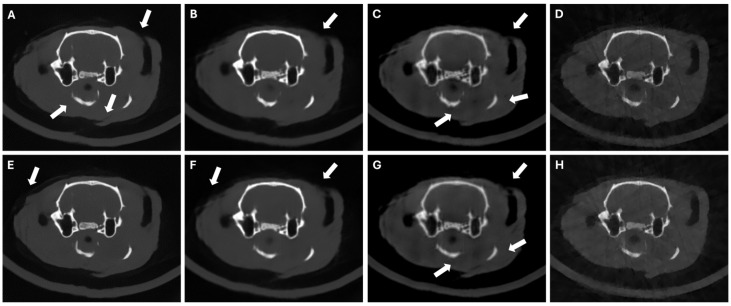
LNP scenario of 42 random projections with random seed = 42 (top) and random seed = 33 (bottom). Axial slices of DeepBH (**A**,**E**), prior images (**B**,**F**), SART-PICCS (**C**,**G**), and PICDL (**D**,**H**). Arrows indicate hallucinations.

**Figure 7 sensors-24-06782-f007:**
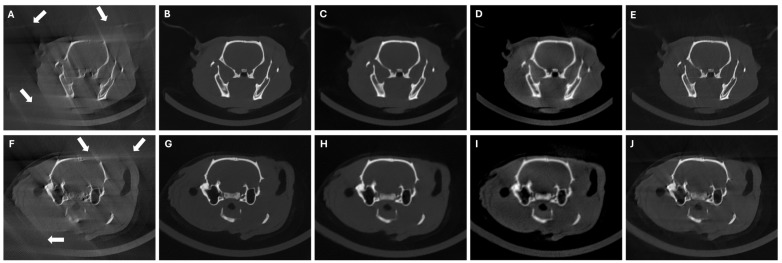
Central axial slices of FDK reconstructions (**A**,**F**), DeepBH reconstructions (**B**,**G**), prior images (**C**,**H**), SART-PICCS reconstructions (**D**,**I**), and PICDL reconstructions (**E**,**J**) for the two test animals in LSA scenario of 120 and 130 projections, respectively. Arrows indicate the LSA artifacts.

**Figure 8 sensors-24-06782-f008:**
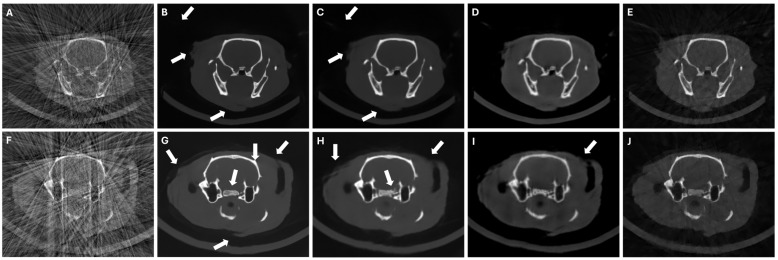
Central axial slices of FDK reconstructions (**A**,**F**), DeepBH reconstructions (**B**,**G**), prior images (**C**,**H**), SART-PICCS reconstructions (**D**,**I**), and PICDL reconstructions (**E**,**J**) for the two test animals in LNP scenario of 49 and 42 projections, respectively. Arrows indicate hallucinations.

**Table 1 sensors-24-06782-t001:** Images used to create the model.

Scenario	SD	LD	LSA	LNP
Training	3361	3361	10,083	10,083
Validation	992	992	2976	2976
Test	992	992	2976	2976

**Table 2 sensors-24-06782-t002:** Regularization parameters used in the PICCS algorithm.

Scenario	Rodent 1			Rodent 2		
	Μ	λ	α	μ	λ	α
LNP	1.6	0.12	0.5	1.4	0.1	0.9
LSA	1.4	0.12	3	1.4	0.12	3

**Table 3 sensors-24-06782-t003:** PSNR, SSIM, and CC calculated for the different LNP random state results.

Random Seed Value	PSNR (dB)			SSIM			CC		
	Prior	SART	PICDL	Prior	SART	PICDL	Prior	SART	PICDL
42	25.82	23.99	32.07	0.50	0.61	0.83	0.93	0.91	0.96
33	25.47	24.41	31.85	0.52	0.63	0.84	0.93	0.91	0.96

**Table 4 sensors-24-06782-t004:** PSNR, SSIM, and CC calculated for the different scenarios and test images.

Metric	Reconstruction Method	Rodent 1		Rodent 2	
		LSA	LNP	LSA	LNPs
PSNR (dB)	FDK	19.81	15.81	15.09	15.52
	Prior	26.45	23.63	27.97	25.82
	SART	23.32	24.66	25.10	23.99
	PICDL	31.37	29.37	30.24	32.07
SSIM	FDK	0.680	0.357	0.512	0.327
	Prior	0.674	0.454	0.738	0.500
	SART	0.733	0.619	0.732	0.610
	PICDL	0.840	0.789	0.856	0.831
CC	FDK	0.793	0.782	0.824	0.722
	Prior	0.968	0.964	0.935	0.925
	SART	0.912	0.933	0.907	0.913
	PICDL	0.961	0.955	0.953	0.960

## Data Availability

The original data presented in the study are openly available in zenodo at https://zenodo.org/records/11952603 (accessed on 1 June 2024).

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
