# Peer review of "Hybrid Reconstruction Approach for Polychromatic Computed Tomography in Highly Limited-Data Scenarios"

_sensors, 2024, doi:10.3390/s24216782_

Round 1
Reviewer 1 Report
Comments and Suggestions for Authors
This paper introduces the Deep Learning-based Prior Image Constrained (PICDL) framework, a hybrid approach designed to produce CT images free from beam-hardening artifacts in various limited data scenarios. The method combines the Prior Image Constrained Compressed Sensing (PICCS) algorithm with a prior generated from an initial FDK reconstruction, utilizing deep learning techniques. The literature review is thorough, and the problem statement is well-articulated. However, I still have some concerns:
1. Why is down sampling and up sampling performed before and after DeepBH, and what type of down sampling and up sampling is used?
2. What’s the n in Eq. 2? Is it energy levels of polychromatic imaging system? How does the author deal with polychromatic imaging system?
3. To demonstrate the necessity of the proposed method, it is essential to compare it with state-of-the-art approaches. For example, [1,2] have worked on metal artifact reduction and sparse-angle CT imaging.
4. Please unify the citation style throughout the manuscript. Currently, there is an inconsistency between citing references as "Zhou Li-ping et al. [10]" and "Ji et al. [11]."
5. Some typos, such as “LSA (E) and LNP (F)” should be “LSA (D) and LNP (E)”.
[1] Zhou B, Chen X, Zhou S K, et al. DuDoDR-Net: Dual-domain data consistent recurrent network for simultaneous sparse view and metal artifact reduction in computed tomography. Medical Image Analysis, 2022.
[2] Wang H, Zhou M, Wei D, et al. Mepnet: A model-driven equivariant proximal network for joint sparse-view reconstruction and metal artifact reduction in ct images. MICCAI, 2023.
Reviewer 2 Report
Comments and Suggestions for Authors
The manuscript presents an interesting hybrid method for reconstruction of 3D CT images of small animals, which allows to compensate for beam hardening artifacts in limited-data conditions. The method combines a priori image generation using a deep learning approach and a compressed-sensing-inspired reconstruction algorithm with original hybrid regularization. The experimental results presented by the authors have proven the effectiveness of the method. The manuscript could be of interest to the reader. Unfortunately, there are some faults (see my comments below) which do not allow me to recommend its publication in Sensors in the form it is submitted. I suggest that the authors thoroughly consider my comments and make a major revision of the manuscript.
Major comments

I would recommend that the authors carefully check the manuscript again for stylistic and grammatical errors. Some sentences are difficult to read and should be simplified or broken into several sentences. Several grammatical errors are also present in the text. Examples:
Round 2
Reviewer 1 Report
Comments and Suggestions for Authors
The concerns were well addressed.
Comments on the Quality of English LanguageNeed to polish.
Author Response
We have done a thorough check of grammar and punctuation.
Reviewer 2 Report
Comments and Suggestions for Authors
The authors have done a good job of improving the quality of their manuscript. In fact, all my comments, including the major ones, have been considered in the most appropriate manner. However the new version of their manuscript is still not free from minor faults (see my comments below) which need to be corrected and after that the manuscript can be published in Sensors.
Comments
1. Line 30: Keywords. In the context of the corrections and designations the authors have made it seems appropriate to replace “PICCS” by “L2-PICCS”.
2. Line 136: Formula 1. In the new version of the manuscript the authors replaced “β” by “1-α” but in the text below the formula (line 139) “β” is present. I think this fault should be corrected.
3. Line 138. “T1 and T2 are the total variation pseudo-norm”. Here one can find an error not only in grammar but also in meaning. In the context of the entire manuscript, T2 is not the total variation norm, but the L2-norm. Am I right?
4. Line 148. “Equation 2 can be solved …”. Seemingly, this is one more fault. Instead of “Equation 2”, there should be “Equation 1”.
5. Line 214. “Experiments and Results” seems to be a title of Section 3. It should not be stuck to the text but stand separately.
6. Lines 371 and 379. Here again it would be better to replace “PICCS” by “L2-PICCS”.
7. Some errors in grammar and faults are still present in the new version of the manuscript. Examples are listed below. That is why I recommend the authors to check once more the correctness of words and sentences, and the use of punctuation marks.
7.1. Line 76. Instead of “None of these works address …” there should be “None of these works addresses …”.
7.2. Lines 138 and 139. Grammatically, instead of “T1 and T2 are the total variation pseudo-norm” there should be “T1 and T2 are the total variation pseudo-norms” but it is not so in the meaning (see my comment 3).
7.3. Line 214. “ResultsWe” – word spacing is needed, or even line spacing (see my comment 5).
7.4. Line 218. “scenarios,either” and “withlimited” also need spacing.
7.5. Line 305. “…scenario of with 42 random …” – either one of the two prepositions is excessive or something is missed between them.
7.6. Line 371. “…allows to solve…” is not correct. There should be “…allows us to solved…” or “…allows the PICCS problem to be accurately solved…”.
7.7. Line 391. Instead of “…allows to correct…” there should be “allows us to correct…” or “…allows beam-hardening to be corrected…”.
Comments on the Quality of English LanguageSome errors in grammar and faults are still present in the new version of the manuscript. Examples are listed below. That is why I recommend the authors to check once more the correctness of words and sentences, and the use of punctuation marks.
1. Line 76. Instead of “None of these works address …” there should be “None of these works addresses …”.
2. Lines 138 and 139. Grammatically, instead of “T1 and T2 are the total variation pseudo-norm” there should be “T1 and T2 are the total variation pseudo-norms” but it is not so in the meaning (see my comment 3).
3. Line 214. “ResultsWe” – word spacing is needed, or even line spacing (see my comment 5).
4. Line 218. “scenarios,either” and “withlimited” also need spacing.
5. Line 305. “…scenario of with 42 random …” – either one of the two prepositions is excessive or something is missed between them.
6. Line 371. “…allows to solve…” is not correct. There should be “…allows us to solved…” or “…allows the PICCS problem to be accurately solved…”.
7. Line 391. Instead of “…allows to correct…” there should be “allows us to correct…” or “…allows beam-hardening to be corrected…”.
Author Response
- Line 30: Keywords. In the context of the corrections and designations the authors have made it seems appropriate to replace “PICCS” by “L2-PICCS”.
Corrected.
- Line 136: Formula 1. In the new version of the manuscript the authors replaced “β” by “1-α” but in the text below the formula (line 139) “β” is present. I think this fault should be corrected.
The “β” has been removed.
- Line 138. “T1and T2 are the total variation pseudo-norm”. Here one can find an error not only in grammar but also in meaning. In the context of the entire manuscript, T2 is not the total variation norm, but the L2-norm. Am I right?
We agree that it was confusing and have rewritten the two paragraphs related to T1 and T2 to try to clarify this point.
- Line 148. “Equation 2 can be solved …”. Seemingly, this is one more fault. Instead of “Equation 2”, there should be “Equation 1”.
Corrected.
- Line 214. “Experiments and Results” seems to be a title of Section 3. It should not be stuck to the text but stand separately.
We have corrected the typo.
- Lines 371 and 379. Here again it would be better to replace “PICCS” by “L2-PICCS”.
Corrected.
- Some errors in grammar and faults are still present in the new version of the manuscript. Examplesare listed below. That is why I recommend the authors to check once more the correctness of words and sentences, and the use of punctuation marks.
We have done a thorough check of grammar and punctuation.
7.1. Line 76. Instead of “None of these works address …” there should be “None of these works addresses …”.
Corrected.
7.2. Lines 138 and 139. Grammatically, instead of “T1 and T2 are the total variation pseudo-norm” there should be “T1 and T2 are the total variation pseudo-norms” but it is not so in the meaning (see my comment 3).
Corrected.
7.3. Line 214. “ResultsWe” – word spacing is needed, or even line spacing (see my comment 5).
Corrected.
7.4. Line 218. “scenarios,either” and “withlimited” also need spacing.
Corrected.
7.5. Line 305. “…scenario of with 42 random …” – either one of the two prepositions is excessive or something is missed between them.
Corrected.
7.6. Line 371. “…allows to solve…” is not correct. There should be “…allows us to solved…” or “…allows the PICCS problem to be accurately solved…”.
Corrected.
7.7. Line 391. Instead of “…allows to correct…” there should be “allows us to correct…” or “…allows beam-hardening to be corrected…”.
Corrected.